# Combining cash transfers and cognitive behavioral therapy to reduce antisocial behavior in young men: A mediation analysis of a randomized controlled trial in Liberia

Marina Xavier Carpena[1], Cristiane Silvestre Paula[2‡], Christian Loret de Mola[3,4,5*], Philipp Hessel[6,7‡], Mauricio Avendano[8‡], Sara Evans-Lacko[9‡], Alicia Matijasevich[10‡]

1 Developmental Disorders Program, Center of Biological Science and Health, Universidade Presbiteriana Mackenzie, São Paulo, Brazil, 2 Programa de Pós-Graduação em Distúrbios do Desenvolvimento e Centro Mackenzie de Pesquisa sobre a Infância e Adolescência- Universidade Presbiteriana Mackenzie (UPM), São Paulo-SP, Brazil, 3 Programa de Pós-Graduação em Saúde Pública, FURG, Rio Grande, RS, Brazil, 4 Grupo de Pesquisa e Inovação em Saúde, Universidade Federal do Rio Grande (FURG), Rio Grande, RS, Brazil, 5 Universidad Cientifica del Sur, Lima, Peru, 6 Swiss Center for Tropical and Public Health, Household Economics and Health Systems Research Unit, Basel, CH, Switzerland, 7 Universidad de los Andes, Alberto Lleras Camargo School of Government, Bogotá, Colombia, 8 Center for Primary Care and Public Health (Unisanté), Department of Epidemiology and Health Systems, University of Lausanne, Lausanne, Switzerland, 9 Care Policy and Evaluation Centre, Department of Health Policy, London School of Economics and Political Science, London, United Kingdom, 10 Departamento de Medicina Preventiva, Faculdade de Medicina FMUSP, Universidade de São Paulo, São Paulo, Brasil

☯ These authors contributed equally to this work.
‡ CSP, PH, MA, SEL, and AM also contributed equally to this work.
* chlmz@yahoo.com

## Abstract

### Background

Interventions that combine cognitive behavioral therapy (CBT) with unconditional cash transfers (UCT) reduce the risk of antisocial behavior (ASB), but the underlying mechanisms are unclear. In this paper, we test the role of psychological and cognitive mechanisms in explaining this effect. We assessed the mediating role of executive function, self-control, and time preferences.

### Methods

We used data from the Sustainable Transformation of Youth in Liberia, a community-based randomized controlled trial of criminally engaged men. The men were randomized into: Group-1: control (n = 237); and Group-2: CBT+UCT (n = 207). ASB was measured 12–13 months after the interventions were completed, and the following mediators were assessed 2–5 weeks later: (i) self-control, (ii) time preferences and (iii) executive functions. We estimated the natural direct effect (NDE) and the natural indirect effect (NIE) of the intervention over ASB.

### Results

Self-control, time preferences and a weighted index of all three mediators were associated with ASB scores, but the intervention influenced time preferences only [B = 0.09 95%CI

**Data Availability Statement:** The data sources for the quantitative analysis are secondary sources,

which are already publicly available at: https://www.openicpsr.org/openicpsr/project/113056/version/V1/view.

**Funding:** This study is an output of the CHANCES-6 study. This work was supported by the UKRI's Global Challenges Research Fund (ES/S001050/1). The support of the Economic and Social Research Council (ESRC) is gratefully acknowledged. CSP received support from CAPES/PRINT grant number 88887.310343/2018-00 and Fundo Mackpesquisa. CSP and SE-L received a grant from Newton funding, grant number: NAFR12\1020; MC, CSP, and SE-L, and received a grant from the Economic and Social Research Council for the CHANCES-6 project (ESRC, ES/S001050/1); CSP and AM have bolsa produtividade em pesquisa from the Brazilian National Council for Scientific and Technological Development (Conselho Nacional de Desenvolvimento Científico e Tecnológico – CNPq). CSP was partially funded by the CAPES Coordenação de Aperfeiçoamento de Pessoal de Nível Superior, Programa Institucional de Internacionalização (CAPES/PrInt), grant number 88887.696831/2022-00. The funders had no role in study design, data collection and analysis, decision to publish, or preparation of the manuscript.

**Competing interests:** The authors have declared that no competing interests exist.

(0.03; 0.15)]. There was no evidence that the effect of the intervention on ASB was mediated by self-control [$B_{NIE}$ = 0.007 95%CI (-0.01; 0.02)], time preferences [$B_{NIE}$ = -0.02 95%CI (-0.05; 0.01)], executive functions [$B_{NIE}$ = 0.002 95%CI (-0.002; 0.006)] or the weighted index of the mediators [$B_{NIE}$ = -0.0005 95%CI (-0.03; 0.02)].

## Conclusions

UCT and CBT lead to improvements in ASB, even in the absence of mediation via psychological and cognitive functions. Findings suggest that the causal mechanisms may involve non-psychological pathways.

## Background

Antisocial behavior (ASB) refers to disruptive acts characterized by covert and overt hostility, intentional aggression, and conduct disorders that often result in criminal or violent behaviors. Conduct disorders and antisocial behavior often precede violence, a leading cause of social instability, injury, mental health problems, and death among young people [1–4]. Violence is considered a preventable problem that has large effects on individuals and society [5–7]. Violence and ASB have harmful consequences for current and future generations, highlighting the need to identify effective interventions [8]

Individuals exposed to poverty experience greater exposure to environments involving violence and crime, which in turn increases the risk of participation in criminal activities [2]. Cash transfer (CT) programs have been implemented in many low-and-middle-income countries as a strategy to increase social protection and reduce poverty, which may also reduce the risk of ASB, crime and violence. CTs supplement the income of poor families, increasing their consumption of food and other basic items. Recent evidence suggests that CTs may also reduce ASB, homicide rates, and even externalizing but not internalizing problems, but effects vary across studies and countries, and the overall evidence of an effect on mental health outcomes is mixed [9–12]

Cognitive behavioral therapy (CBT) refers to a class of interventions that share the basic premise that mental disorders and psychological distress are maintained by cognitive factors [13]. The core premise proposes that maladaptive cognition contributes to the maintenance of emotional distress and behavioral problems. Therapeutic CBT strategies focus on changing these maladaptive cognitions to promote changes in emotional distress and problematic behaviors [14]. CBT has been used for several mental health conditions and problems, including depression, anxiety, somatoform disorders, bulimia, anger control problems, and general stress [13], and has also been proposed as a therapy for ASB [15]. However, there is only weak evidence that any psychological treatment (including CBT) reduces antisocial personality disorder, and from the few studies addressing antisocial behavior/disorder available, most have been conducted in high-income countries [16].

The Sustainable Transformation of Youth in Liberia (STYL) trial was the first experimental study to evaluate the impact of combining unconditional cash transfers (UCTs) and CBT on ASB, relative to each intervention alone. This study was carried out in young men in Liberia and is unique in that it used a randomized control trial (RCT) design [10]. One of the key findings of this study was that neither CBT nor UCT alone influence ASB 12–13 months post intervention, while the combination of CBT and UCT leads to significant reductions in ASB. However, the causal mechanisms by which the combination of UCT and CBT influence ASB

remains unknown. Elucidating causal mechanisms is important to design better interventions that target the most important channels through which ASB influences outcomes and how they may be influenced [17]. A causal mediation may also point to key components that might be most useful to incorporate in future interventions [18]. If we could determine which parameters change and consequently produce the improvement expected by the intervention, we could focus efforts or rethink interventions to make them more efficient. Potential explanations of the effect of UCT and CBT on ASB involve two competing hypotheses: on the one hand, the treatment may generate individual changes in psychological and cognitive functions, which may underlie changes in ASB. For example, the intervention may increase the ability to control emotions, thus leading to reductions in ASB. A second hypothesis, however, suggests that changes in cognitive psychological outcomes are not a requirement for the intervention effect. Instead, changes may be due to other factors, such as changes in the social environment in which youths live, including reductions in exposure to violent and crime environments and changes in social networks.

In their original analysis, Blattman *et al.* [10] found that only the group that received both CBT and UCT experienced a reduction in ASB. For this study we hypothesis that executive function, self-control, and time preferences are a plausible mediator of the effect of combined CBT and UCT on ASB. This hypothesis is supported by previous research showing that higher executive function, self-control, and time preferences are negatively associated with ASB [4] and involvement in criminal activities [19]. We exploit the unique setting of the STYL study in Liberia, which included extensive measures of executive function, self-control, and time preferences, and two longitudinal measures which enable assessment of their potential mediation role.

Therefore, this study is an extension of the original analysis and aims to evaluate the potential psychological mechanisms by which an intervention that combines UCT with CBT influences ASB.

## Materials and methods

### Sample and study design

We used the publicly available dataset of the STYL trial [10]. This was a community based randomized trial using a 2 × 2 factorial design. A total of 999 criminally engaged Liberian men were randomized into four groups using neighborhood weights (sample stages and a brief design description are shown in Fig 1). The experimental arms included four groups: (1) No intervention (i.e., waiting list control group); (2) Eight weeks of CBT, focused on self-regulation, patience, and noncriminal values; (3) Lottery for a $200 grant, about three months' wages (UCT); and (4) Both CBT and UCT.

To assess our hypothesis, we only included data from Group-1: control (n = 207); and Group-4: CBT+UCT (n = 237). The original flow of participants for this intervention study is shown in Fig 1.

### Variables

Our study involves a comparison between two arms: the CBT plus UCT (CBT+UCT group) versus no intervention (control group). We used the ASB index developed by Blattman *et al.* as the main outcome. The ASB index is a standardized index that includes self-reported drug selling, stealing, interpersonal fighting, weapons carrying, arrests, hostile attitudes, and domestic abuse. It was created based on these seven measures from sets of related survey questions (self-reported data) relating to disruptive or harmful acts toward others, such as crime or aggression

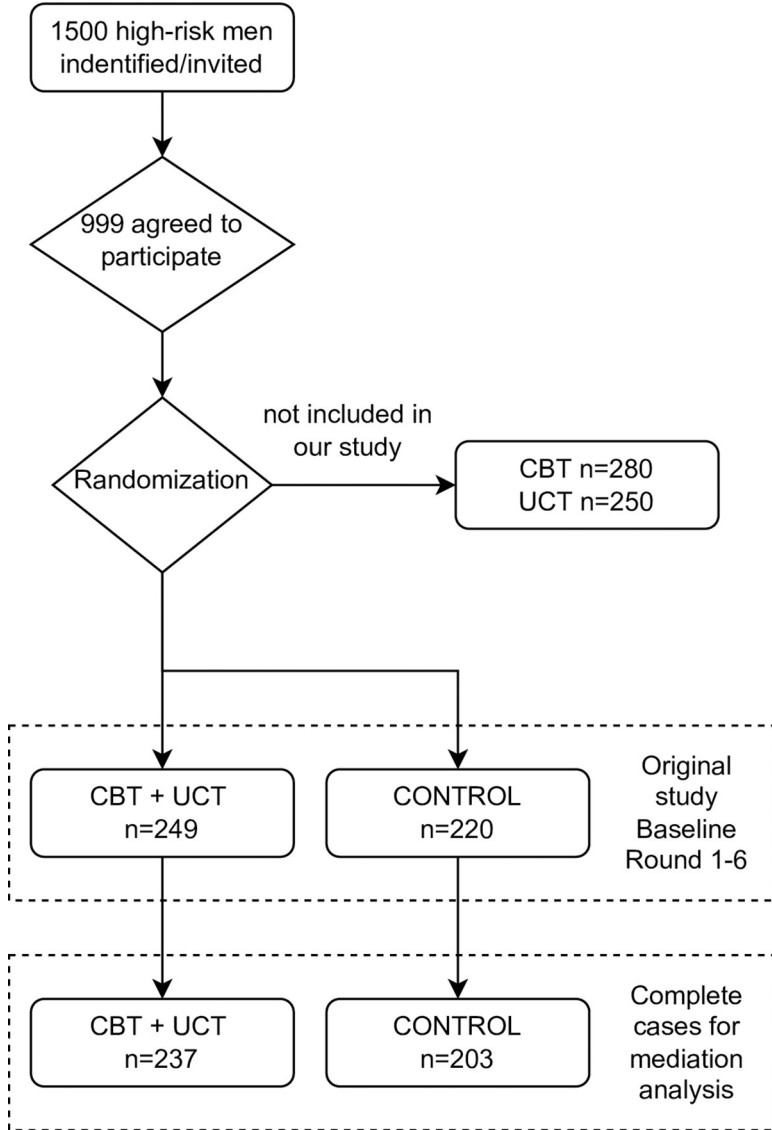

**Fig 1. Flow chart of trial design and those included in the analysis.** Open acces available information can be found at: https://www.nber.org/system/files/working_papers/w21204/w21204.pdf.

evaluated 12–13 months after the intervention was finished (rounds 5 and 6 presented in S1 Fig). More information is available in the original publication [10].

We assessed four possible mediators separately 2–5 weeks after the interventions were completed. These included indices of: (i) self-control, (ii) time preferences, (iii) executive function, and (iv) a variable combining all three mediators into one continuous index. The self-control variable was created using a series of Likert scale items exploring impulsiveness (N = 8), conscientiousness (N = 8), perseverance (N = 7) and reward responsiveness (N = 8) (a description of each subscale is available as supporting material). Time preferences assess the importance or value a person gives to receiving a good or cash at an earlier date compared with receiving it later, e.g., whether a person would prefer to receive a smaller amount of money now compared to a larger amount of money more in the future. We used a summary index of eight equally

weighted components: four measures for patience and four measures for time inconsistency. A lower score indicates a preference to receive a good or cash earlier.

Executive function was evaluated using three interactive activities drawn from economics and psychology, including: planning behavior, behavior inhibition and cognitive flexibility, and working memory. The overall summary index for each mediator is the standardized mean of its composite item. A global index was created by combining each mediator index into an equally weighted index of self-control, time preferences, and executive function (SCTPEF).

Finally, as covariates we used the following baseline measures related to socioeconomic and health status: age, living with a partner, living with a person under 15 years, schooling (measured as the total number of years of formal education), having a disability, if the participant ever sold drugs, current drug use, and a mental health z-score index (i.e., an index based on several questions to evaluate depression and distress symptoms, neuroticism, self-esteem, and locus of control).

## Statistical analysis

We tested the following null hypothesis, presented in Fig 2: self-control, time preferences, and executive function do not mediate the effect of CBT plus UCT on ASB. The alternative hypothesis is that at least one of these dimensions is a mediator of the effect of CBT plus UCT on ASB. We estimated the natural direct effect (NDE), natural indirect effect (NIE) and controlled direct effect (CDE), of each mediator on ASB, using each mediator separately [20]. We used an intention to treat (ITT) analysis.

The NDE represents the effect of exposure on the outcome that is not mediated by the putative mediator, while the NIE corresponds to the effect that is mediated by the putative mediator. The sum of the NDE and NIE represents the total casual effect, and the quotient of dividing the NIE by the total effect represents the percentage of the effect that is mediated by the putative mediator. The CDE represents the effect of the exposure on the outcome if the mediator could be controlled (maintained constant or fixed at one level). To calculate this, we used the "paramed" module of the STATA v.13.1 program. Standard errors for mediation analyses were calculated using bootstrapping with 5000 simulations. The Benjamini and Hochberg False Discovery Rate (BH-FDR) was used to correct for multiple testing [21, 22]

The original study by Blattman *et al.* had two endpoints, for which outcomes and potential mediators were assessed: a short-term assessment 2–5 weeks after the intervention was completed and a long-term assessment at 12–13 months (Fig 1). For this study we considered outcomes assessed at 12–13 months, and mediators assessed in the short-term evaluation, in line with the timing of a causal framework [20].

In addition, we used multiple regression models to assess the association between the intervention and the mediators, as well as the mediators and the outcome of ASB, adjusted for the previously mentioned covariates.

All mediators and outcome variables were standardized in Z-scores. Therefore, interpretation of results for betas in all models should be in standard deviation (SD).

Considering an alpha of 0.05, we had a power of 80%., we calculated the power needed with the sample size to compare the groups and find mean differences of at least 0.05 (SD = 0.1) for the standardized index scores of ASB.

## Ethics

The original study was approved by the Institutional Review Boards of the University of Liberia. IRB Approval Number: IRB19-0961-AM008. Participants proved provided informed to participate. Ethics approval for secondary analysis of quantitative data conducted at King's

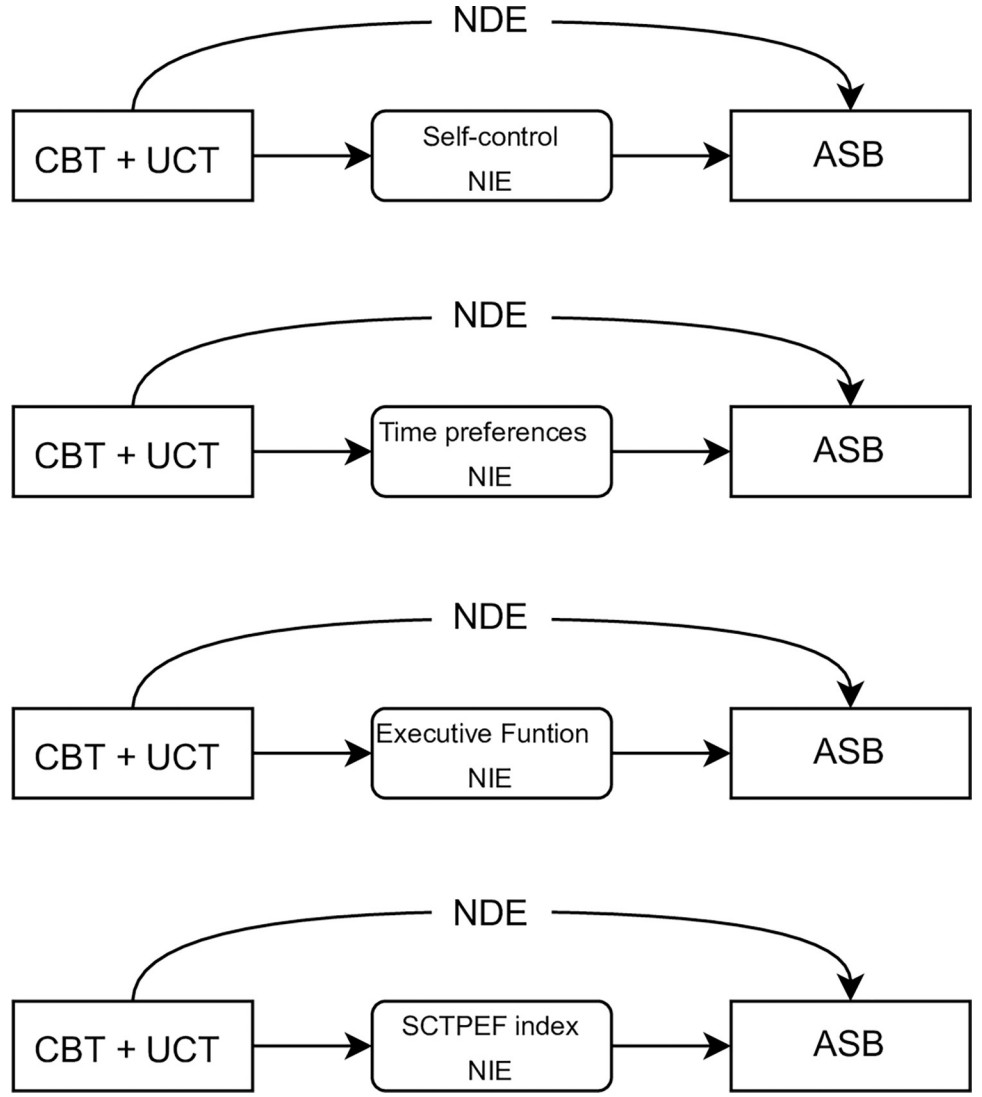

**Fig 2. Direct acyclic graphs, of the STYL experiment on antisocial behavior, and hypothesized mediators.** NDE, natural direct effect; NIE, natural indirect effect; SCTPEF: self-control time preferences and executive function unique index. All of these refer to the mediating effect on the association between Group CBT + UCT vs. control group over Antisocial Behavior (ASB).

College London was obtained from the Ethics Office of King's College London Research (LRS-19/20-15050). Only persons who provided informed consent for the original study were included in the study.

## Results

We used data of 237individuals in the CBT + UCT group and 203from the control for whom information for both the mediator and outcome (ASB) variables was available. Originally, we had in the CBT + UCT group 249 individuals and 220 in the control group (Fig 1), which represent 5% and 8% of missing data in each group, respectively. The overall sample had a mean age of 25.36 years (SD = 4.83) and mean schooling of 7.73 years (SD = 3.30). About 16% were living with a partner whereas almost 49.1% were living with someone under 15. Almost two

**Table 1. Sample description according to covariates assessed at baseline.**

|  | Groups | | |
|  | CBT + UCT N = 237 | Control N = 203 | p-value |
|---|---|---|---|
| Age (years) | 25.25 (4.7) | 25.34 (5.0) | 0.965* |
| Living with partner (yes) | 39 (15.7) | 40 (18.2) | 0.717** |
| Living with a <15-year-old (yes) | 117 (49.4) | 105 (51.7) | 0.257** |
| Schooling (number of years) | 7.76 (3.42) | 7.82 (2.37) | 0.9314* |
| Ever sold drugs (yes) | 52 (21.9) | 38 (18.7) | 0.755** |
| Drug user (yes) | 140 (59.1) | 113 (55.7) | 0.237** |
| Sleeping on the streets (yes) | 53 (22.4) | 49 (24.1) | 0.936** |
| Mental health z score | -0.03 (0.98) | 0.002 (0.96) | 0.919** |

* P-value for the ANOVA

** P-value for the qui-squared test

CBT: cognitive behavior therapy; UCT: unconditional cash transfer

thirds reported that they often smoked grass or took hard drugs (60.8%), and 19.5% reported often selling drugs. Table 1 shows that there were no significant differences between treatment conditions and controls across baseline characteristics.

Multiple regression models showed that self-control [B = -0.24 95%confidence interval [CI] (-0.31; -0.17)], time preferences [B = -0.26 95%CI (-0.32; -0.20)], and the weighted index of mediators [B = -0.21 95%CI (-0.31; -0.11)], were associated with ASB scores, but executive function was not [B = -0.21 95%CI (-0.08; 0.04)]. When we analyzed if the intervention influenced the mediators, ITT analysis only showed an effect for time preferences [B = 0.09 95%CI (0.03; 0.15)].

Table 2 shows our mediation models. Total effect scores showed that the CBT + UCT group presented a reduction in total ASB scores in the long-term evaluations. However, there was no evidence that the effect of the intervention on ASB was mediated by self-control [$B_{NIE}$ = 0.007 95%CI (-0.01; 0.02)], time preferences [$B_{NIE}$ = -0.02 95%CI (-0.05; 0.01)], executive functions [$B_{NIE}$ = 0.002 95%CI (-0.002; 0.006)] or by the weighted index of the mediators (SCTPEF) [$B_{NIE}$ = -0.0005 95%CI (-0.03; 0.02)].

## Discussion

### Main findings

Our aim was to test whether changes in psychological and cognitive functions explain the effect of an intervention that combines UCT and CBT on ASB. There was a positive effect of

**Table 2. Total, natural direct (NDE) and indirect effect (NIE) of the STYL experiment on antisocial behavior using intention to treat (ITT) Z-score estimates.**

|  | NDE | NIE | NIE | NIE | TOTAL EFFECT |
|  | Beta (95%CI) | Beta (95%CI) | p-value | FDR-adjusted p-value | Beta (95%CI) |
|---|---|---|---|---|---|
| Self-control | -0.31 (-0.45; -0.18) | 0.007 (-0.01; 0.02) | 0.405 | 0.653 | -0.31 (-0.46; -0.15) |
| Time preferences | -0.26 (-0.45; -0.08) | -0.02 (-0.05; 0.01) | 0.195 | 0.585 | -0.28 (-0.46; -0.10) |
| Executive Funtions | -0.25 (-0.35; -0.14) | 0.002 (-0.002; 0.006) | 0.332 | 0.653 | -0.24 (-0.34; -0.15) |
| SCTPEF index | -0.26 (-0.44; -0.08) | -0.0005 (-0.03; 0.02) | 0.414 | 0.653 | -0.27 (-0.45; -0.09) |

Note: NDE, natural direct effect; NIE, natural indirect effect; SCTPEF: self-control time preferences and executive function unique index; CI: confidence interval; FDR: false discovery rate. All of these refer to the mediating effect on the association between Group CBT + UCT vs. control group using ITT. Adjusted for age, living with partner, living with an under of 15 years old, schooling, disabled, drugs sell ever, drug user, mental health z score, all evaluated at baseline.

CT and CBT on time preferences, but none of the cognitive functions mediated the effect of the intervention on ASB. Our results suggest that the effect of the combination of CBT with UCT on ASB does not seem to be caused by improved executive function, self-control, or time preferences, and raise important questions about what other mechanisms may be at play to explain the strong effect of the intervention on ASB.

## Comparison with prior literature

To our knowledge, no prior study has empirically examined the mediating factors that explain the effect of combining UCT and CBT interventions on ASB. Previous studies have found little evidence that psychological skills including emotional intelligence or social skills, self-control or "grit," mediate the effect of behavioral interventions on crime [23–25]. This raises the question about the role of other mechanisms by which CBT or UT may affect ASB outcomes. Existing theoretical frameworks suggest that cash transfers might reduce ASB by inducing positive changes in drug abuse, social networks, and mental health outcomes (Lund et al 2018), mechanisms that may also apply for CBT interventions [14]. In their original study, Blattman *et al.* (2017) suggested that time preference, self-control and executive changes might be some of the hypothesized channels by which CBT and UCT reduce ASB. There is a theoretical basis for this: it is often hypothesized that youth programs help individuals to have better introspection of their automatic thoughts and behaviors, how they face each situation, and how the situation could be construed differently [25].

Evidence from our study suggests that CBT combined with UCT did not generate meaningful changes in key psychological and cognitive functions, which is consistent with the finding that CBT alone does not reduce ASB. Possible mechanisms may include changes in behavior such as drug abuse and conduct problems; psychological and cognitive functions other than those assessed in our study, such as resilience, temperament, or coping mechanisms; family dynamics, rearing practices, or social networks; and changes in the social environment [26].

## Strengths and limitations

This is the first study to investigate mechanisms explaining the effect of CBT and UCT on ASB using counterfactual outcome mediation analysis [27]. We analyzed a unique, and methodologically robust dataset which applied a 2x2 factorial design conducted in a hard-to-reach population, incorporating a high number of variables and relevant information to control the analysis for confounders, as well as very few missing data (5.6%). Recently, the authors of the original study reported that the effect of CBT plus CT was maintained across time, highlighting that the combination of these interventions led to roughly 34 fewer thefts and robberies per year per subject at both the 1- and 10-year evaluations [10]. Our counterfactual-based analyses can account for post-treatment confounders, which affect only the mediator and outcome in the model and can be affected by the exposure/intervention (intermediate confounder). Due to random treatment assignment, in an RCT we need not be concerned with confounders of the treatment-outcome or treatment-mediator relationship. However, in a mediation model, we need to calculate the total effect size, the effect size of the exposure on the mediator, and the effect of the mediator on the outcome [20].

Our study, however, has some important limitations. First, we cannot be sure that our estimates of the relationship between mediators and ASB is causal. Although we controlled for observables, the fact that mediators are not randomly assigned means that confounding may still bias our estimates of mediation of the effect of treatment on ASB. An alternative approach would be to use an instrumental variable approach that exploits potentially exogenous variation in mediators and incorporate these in the models. However, identifying valid instruments

for executive function, self-control, and time preferences is challenging. Future studies should examine to what extent the relationship between executive function, self-control, and time preferences, and ASB is causal.

We had some losses to follow-up, which are most likely not at random, this could affect our measures of association. Nonetheless, missing data was less than 8%, and we consider that this would not have had a great impact in our overall analyses. We had to limit our study to a subset of mechanisms, i.e., executive function, self-control, and time preferences. These involve malleable skills that can be trained or even improved by therapy or mental training [28, 29]. However, there might be other psychological mechanisms which could be involved in the casual pathway that were not available for testing in this database, such as resilience, temperament, or coping mechanisms. The original study was not designed to evaluate mediating effects and was not initially powered to test the presence of possible mediators. However, we calculated the power needed with the sample size and since our calculations showed that we were able to find at least differences of 0.05 and we consider that the lack of association found is consistent Finally, our study is based on a population of young men in Liberia living in a high-poverty context and with high exposure to violence and crime. Executive function, self-control, and time preferences may be difficult to change in this context, or the effect of these on ASB may be less important within a social context in which violence and crime are the norm. This may explain the fact that these variables did not explain the effect of the treatment on ASB in Liberia. Future studies should examine whether executive function, self-control, and time preferences can explain a more important share of the effect of UCTs and CBT on ASB in less deprived and socially challenging contexts.

## Conclusion

A combination of UCTs and CBT reduced the risk of ASB in Liberia. Improvements in executive function, self-control and time preferences were hypothesized as explanations of this effect. CT and CBT improved time preferences, but it had no effect on self-control or executive function. Overall, none of these factors was in the causal pathway between the intervention and reductions in ASB. Our findings highlight the need to identify alternative mechanisms by which CTs and CBT affect ASB. While this may include changes in other cognitive functions, our results suggest that part of the explanation may lie in changes in other factors such behavior and conduct problems; psychological and cognitive functions other than those assessed in our study; family dynamics, rearing practices or social networks; and changes in the social environment [26]. Future studies should examine whether the intervention led to changes in some of these factors, including reduced exposure to violence and crime environments, and changes in social networks that may have reinforced positive behaviors.

## Supporting information

**S1 Fig. Flow chart of trial design.** Blue shows those who were included in the current analysis.
(TIF)

## Author Contributions

**Conceptualization:** Marina Xavier Carpena, Mauricio Avendano, Alicia Matijasevich.

**Data curation:** Marina Xavier Carpena, Christian Loret de Mola.

**Formal analysis:** Marina Xavier Carpena, Christian Loret de Mola.

**Funding acquisition:** Cristiane Silvestre Paula, Alicia Matijasevich.

**Investigation:** Marina Xavier Carpena, Cristiane Silvestre Paula, Philipp Hessel, Alicia Matijasevich.

**Methodology:** Marina Xavier Carpena, Cristiane Silvestre Paula, Christian Loret de Mola, Philipp Hessel, Alicia Matijasevich.

**Project administration:** Cristiane Silvestre Paula, Alicia Matijasevich.

**Resources:** Cristiane Silvestre Paula, Mauricio Avendano, Sara Evans-Lacko, Alicia Matijasevich.

**Supervision:** Marina Xavier Carpena, Cristiane Silvestre Paula, Alicia Matijasevich.

**Validation:** Marina Xavier Carpena, Philipp Hessel, Mauricio Avendano, Sara Evans-Lacko, Alicia Matijasevich.

**Visualization:** Marina Xavier Carpena, Philipp Hessel, Mauricio Avendano, Sara Evans-Lacko, Alicia Matijasevich.

**Writing – original draft:** Marina Xavier Carpena, Christian Loret de Mola, Alicia Matijasevich.

**Writing – review & editing:** Marina Xavier Carpena, Cristiane Silvestre Paula, Christian Loret de Mola, Philipp Hessel, Mauricio Avendano, Sara Evans-Lacko, Alicia Matijasevich.

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
