## [Decision Letter · Decision Letter 0]

9 Jan 2023

PONE-D-22-23021Combining cash transfers and cognitive behavioral therapy to reduce antisocial behavior in young men: A mediation analysis of a randomized controlled trial in LiberiaPLOS ONE

Dear Dr. Christian Loret de Mola,

Thank you for submitting your manuscript to PLOS ONE. After careful consideration, we feel that it has merit but does not fully meet PLOS ONE’s publication criteria as it currently stands. Therefore, we invite you to submit a revised version of the manuscript that addresses the points raised during the review process.

We look forward to receiving your revised manuscript.

Kind regards,

Bárbara Oliván-Blázquez, Ph.D.

Academic Editor

PLOS ONE

Journal Requirements:

2. Please provide additional details regarding ethical approval in the body of your manuscript. In the Methods section, please ensure that you have specified the name of the IRB/ethics committee that approved your study.

This study is an output of the CHANCES-6 study. This work was supported by the UKRI’s Global Challenges Research Fund (ES/S001050/1). The support of the Economic and Social Research Council (ESRC) is gratefully acknowledged. AM received support from the National Council for Scientific and Technological Development (CNPq), Brazil. CSP received support from CAPES/PRINT grant number 88887.310343/2018-00 and Fundo Mackpesquisa.

none

6. Please provide additional details regarding participant consent. In the ethics statement in the Methods and online submission information, please ensure that you have specified (1) whether consent was informed and (2) what type you obtained (for instance, written or verbal, and if verbal, how it was documented and witnessed). If your study included minors, state whether you obtained consent from parents or guardians. If the need for consent was waived by the ethics committee, please include this information.

8. We note you have included a table to which you do not refer in the text of your manuscript. Please ensure that you refer to Table 2 in your text; if accepted, production will need this reference to link the reader to the Table.

Additional Editor Comments:

Dear authors,

Thank you for submitting your manuscript to PLOS ONE. After careful consideration, we feel that it does not fully meet PLOS ONE’s publication criteria as it currently stands. Therefore, we invite you to submit a revised version of the manuscript that addresses the points raised during the review process.

The manuscript has been evaluated by two reviewers, and their comments are available below. Could you please revise the manuscript to carefully address the concerns raised?

Thank you very much.

Reviewers' comments:

Reviewer's Responses to Questions

**Comments to the Author**

1. Is the manuscript technically sound, and do the data support the conclusions?

Reviewer #1: Partly

Reviewer #2: No

2. Has the statistical analysis been performed appropriately and rigorously? 

Reviewer #1: Yes

Reviewer #2: Yes

3. Have the authors made all data underlying the findings in their manuscript fully available?

Reviewer #1: Yes

Reviewer #2: No

4. Is the manuscript presented in an intelligible fashion and written in standard English?

Reviewer #1: Yes

Reviewer #2: Yes

5. Review Comments to the Author

Reviewer #1: In this manuscript, the authors test an intervention that combines cognitive behavioral therapy (CBT) with unconditional cash transfers (UCT) to reduce the risk of antisocial behavior (ASB). As the underlying mechanisms are unclear, the authors want to check the role of psychological and cognitive mechanisms to see if explaining this effect. Specifically, they assessed the mediating role of executive function, self-control, and time preferences. In the end it is concluded that UCT and CBT lead to improvements in ASB, even in the absence of mediation via psychological and cognitive functions. So, findings suggest that the causal mechanisms may involve non-psychological pathways.

Please consider the below comments to strengthen the study as reported, which reflect my responses to the checklist below regarding the literature review and methods:

Introduction:

1. Maybe more references for the following sentence are needed: “However, effects vary acros studies and countries, and the overall evidence of an effect on mental health outcomes is mixed. [16]”

2. Be careful with typos: “acros” in the background.

3. The following sentence sounds a bit repetitive to me, rewrite it if possible: “In particular, we examine the hypothesis that executive function, self-control, and time preferences shown to be influenced by the STYL intervention, including executive function, self-control, and time preferences [7],”.

Methods:

1. Figure 1. The reasons why there were losses in each evaluation are not collected. Do you know the authors to what could be due? Put in limitations, as this may produce bias.

2. It is still better to put the total number of participants of the 2 groups that participated in this study, in addition to putting the total that appears in the STYL trial database (n=999).

3. Why is it an intention to treat (ITT) analysis? How have the losses been treated? Has there been data imputation?

Results:

1. I think the equal should be outside the parentheses: “(standard deviation [SD=]4.83)”.

2. Table 1. Is this p-value in Schooling possible?: 9314*

3. Table 2 comes out cut and cannot be seen in its entirety.

4. In addition to talking about indirect and direct effects, which I think is well explained, the authors could consider showing the results in a small graph, where the mediation model could be seen. This would be useful for readers not used to these models. Visually seeing what the indirect and direct effect refers to, I think it could be of great help to improve the quality of the manuscript.

Discussion

1. I think there is a point left over in this sentence: “but none of the cognitive functions .assessed mediated the effect of the intervention on ASB”.

2. If there is a lack of space in the manuscript, I would consider deleting this paragraph or rewriting it: "Whittle et al. [22] extended the findings of Jellema et al. [31] by investigating mediation of the effect of a psychosocial intervention with usual general practitioner care in a primary care population with (sub)acute low back pain [22] on disability outcomes. They tested key factors, such as pain catastrophizing, fear-avoidance beliefs, and distress, Originally proposed by Jellema et al. [31] which, in theory, would be important mediators for their RCT, and found that none of these factors was part of the causal pathway. While this RCT focus on a different question than ours, both Jellema et al. [31] and our paper illustrate how mediation analysis can help us identify the factors that may explain why interventions work and provide useful information about how to redesign or improve them." Simply explaining in the methods section what mediation models are for (and making a graph with the results) would be more than enough for the reader to become familiar with them.

3. It is good to add that certain calculations have been made regarding the sample size ("However, we calculated the power needed with the sample size to compare the groups and find mean differences of at least 0.05 (SD=0.1) for the standardized index scores of ASB. Considering an alpha of 0.05, we had a power of 80%."), despite the fact that the initial RCT used its own. But I think that this information, if you want to add it, should be found in the methods part. Another option would be to ignore this data and say that it is a "secondary data analysis" of an RCT and that therefore, being exploratory in nature, it does not have its own sample calculation.

Reviewer #2: The present study is an extension of a original analysis and aims to evaluate the potential psychological mechanisms by which an intervention that combines unconditional cash transfers (UCT) with cognitive behavioral therapy (CBT) influences antosocial behaviour (ASB). In particular, we examine the hypothesis that executive function, self-control, and time preferences shown to be influenced by the STYL intervention, including executive function, self-control, and time preferences, are a plausible mediator of the effect of combined CBT and UCT on ASB. The specific aim of this paper is to test the role of psychological and cognitive mechanisms in explaining the reduce od the risk of antisocial behavior.

This is a very specific topic that might be more likely to be published in a more specialized journal

The original trial (Blattman C, et al. Reducing crime and violence: Experimental evidence from cognitive behavioral therapy in Liberia. Am Econ Rev. 2017;107: 1165–1206.) is not available in open acces. The methodology is not described but is directed to the original article by Battman, of which there is no free access, so the reader of plos one cannot access important information for the follow-up of this article.

The only figure that shows the manuscript is a copy of the original article with the 4 arms of the essay. it is not clear if there has been a subsequent intervention with only the two arms of the trial used or not...not detailed at all. There is no evidence of losses or dropouts... wasn't there?.

A figure should be detailed with the intervention specific to the two arms of the present study.

In the discussion, very poor in the depth of the analysis, the results are compared with another study in which a similar intervention was used in patients with chronic low back pain. Despite the fact that the same authors explain it, CBT is not really comparable in both groups.

The bibliography should be reviewed, there are citations that do not meet the standards.

6. PLOS authors have the option to publish the peer review history of their article (what does this mean?). If published, this will include your full peer review and any attached files.

Reviewer #1: **Yes: **Alejandra Aguilar-Latorre

Reviewer #2: No

---

## [Author Response · Author response to Decision Letter 0]

2 Feb 2023

Reviewer #1: In this manuscript, the authors test an intervention that combines cognitive behavioral therapy (CBT) with unconditional cash transfers (UCT) to reduce the risk of antisocial behavior (ASB). As the underlying mechanisms are unclear, the authors want to check the role of psychological and cognitive mechanisms to see if explaining this effect. Specifically, they assessed the mediating role of executive function, self-control, and time preferences. In the end it is concluded that UCT and CBT lead to improvements in ASB, even in the absence of mediation via psychological and cognitive functions. So, findings suggest that the causal mechanisms may involve non-psychological pathways.

Please consider the below comments to strengthen the study as reported, which reflect my responses to the checklist below regarding the literature review and methods:

Introduction:

1. Maybe more references for the following sentence are needed: “However, effects vary acros studies and countries, and the overall evidence of an effect on mental health outcomes is mixed. [16]”

Thank you for the comment. We have rephrased the last sentences of that paragraph, and now the statement has more references. It reads:

“…CTs supplement the income of poor families, increasing their consumption of food and other basic items. Recent evidence suggests that CTs may also reduce ASB, depression, anxiety, and homicide rates, and even externalizing but not internalizing problems, but effects vary across studies and countries, and the overall evidence of an effect on mental health outcomes is mixed. [6–12]. “ 

2. Be careful with typos: “acros” in the background.

Thank you for the comment, we have checked the entire document for typos and other probable errors.

3. The following sentence sounds a bit repetitive to me, rewrite it if possible: “In particular, we examine the hypothesis that executive function, self-control, and time preferences shown to be influenced by the STYL intervention, including executive function, self-control, and time preferences [7],”.

We agree with the comments and have rephrased the final paragraph of the introduction. It now reads:

“In their original analysis, Blattman et a.l [7] found that only the group that received both CBT and UCT experienced a reduction in ASB. For this study we hypothesis that executive function, self-control, and time preferences are a plausible mediator of the effect of combined CBT and UCT on ASB. This hypothesis is supported by previous research showing that higher executive function, self-control, and time preferences are negatively associated with ASB [21] and involvement in criminal activities [22]. We exploit the unique setting of the STYL study in Liberia, which included extensive measures of executive function, self-control, and time preferences, and two longitudinal measures which enable assessment of their potential mediation role”

Methods:

1. Figure 1. The reasons why there were losses in each evaluation are not collected. Do you know the authors to what could be due? Put in limitations, as this may produce bias.

Thank you for the comments in fact, that is not explicitly described in the original work. Therefore, we have included it as a limitation of the study. Since losses could have been not at random. We have also edited figure 1 so it is more accurate for our own analysis.

2. It is still better to put the total number of participants of the 2 groups that participated in this study, in addition to putting the total that appears in the STYL trial database (n=999).

Thank you for the comment we have included the numbers as suggested in the results section and figure 1.

3. Why is it an intention to treat (ITT) analysis? How have the losses been treated? Has there been data imputation?

Thank you for the comment. This was an intention to treat analysis, since we consider to be part of the control and treated group those who were randomized to each group in the original study. Therefore, we are testing the randomization as our main exposure. Losses to follow-up were excluded, this is a complete case analysis. We could have imputed the data using multiple imputation models, however using this type of mediation analysis and imputing data could bring some problems to the model and might not converge. And since the losses were so low we preferred not too complicate the model. 

We have been more explicit in the matter of missing data and have included the following statement in the results:

“We used data of all 237 individuals in the CBT + UCT group and 203 from the control for whom information for both the mediator and outcome (ASB) variables was available. Originally, we had in the CBT + UCT group 280 individuals and 220 in the control group (figure 1), which represent 15% and 8% of missing data in each group, respectively.”

Results:

1. I think the equal should be outside the parentheses: “(standard deviation [SD=]4.83)”.

Thank you for the comment we have corrected this.

2. Table 1. Is this p-value in Schooling possible?: 9314*

Thank you for the comment. This was another typo; we have corrected now. it should be 0.9314

3. Table 2 comes out cut and cannot be seen in its entirety.

Thank you, we have arranged this, and is now within the parameters of the manuscript.

4. In addition to talking about indirect and direct effects, which I think is well explained, the authors could consider showing the results in a small graph, where the mediation model could be seen. This would be useful for readers not used to these models. Visually seeing what the indirect and direct effect refers to, I think it could be of great help to improve the quality of the manuscript.

Thank you for the suggestion. We agree that would be helpful and have decided to include a figure 2 showing our hypothesis. 

Discussion

1. I think there is a point left over in this sentence: “but none of the cognitive functions .assessed mediated the effect of the intervention on ASB”.

Thank you for the comment we have corrected it.

2. If there is a lack of space in the manuscript, I would consider deleting this paragraph or rewriting it: "Whittle et al. [22] extended the findings of Jellema et al. [31] by investigating mediation of the effect of a psychosocial intervention with usual general practitioner care in a primary care population with (sub)acute low back pain [22] on disability outcomes. They tested key factors, such as pain catastrophizing, fear-avoidance beliefs, and distress, Originally proposed by Jellema et al. [31] which, in theory, would be important mediators for their RCT, and found that none of these factors was part of the causal pathway. While this RCT focus on a different question than ours, both Jellema et al. [31] and our paper illustrate how mediation analysis can help us identify the factors that may explain why interventions work and provide useful information about how to redesign or improve them." Simply explaining in the methods section what mediation models are for (and making a graph with the results) would be more than enough for the reader to become familiar with them.

Thank you for the comment, we have deleted this paragraph. We agree it was not necessary.

3. It is good to add that certain calculations have been made regarding the sample size ("However, we calculated the power needed with the sample size to compare the groups and find mean differences of at least 0.05 (SD=0.1) for the standardized index scores of ASB. Considering an alpha of 0.05, we had a power of 80%."), despite the fact that the initial RCT used its own. But I think that this information, if you want to add it, should be found in the methods part. Another option would be to ignore this data and say that it is a "secondary data analysis" of an RCT and that therefore, being exploratory in nature, it does not have its own sample calculation.

We agree with the reviewer, we have moved this explanation to the methods section. But also explained that this is a secondary analysis.

Reviewer #2: The present study is an extension of a original analysis and aims to evaluate the potential psychological mechanisms by which an intervention that combines unconditional cash transfers (UCT) with cognitive behavioral therapy (CBT) influences antosocial behaviour (ASB). In particular, we examine the hypothesis that executive function, self-control, and time preferences shown to be influenced by the STYL intervention, including executive function, self-control, and time preferences, are a plausible mediator of the effect of combined CBT and UCT on ASB. The specific aim of this paper is to test the role of psychological and cognitive mechanisms in explaining the reduce od the risk of antisocial behavior.

This is a very specific topic that might be more likely to be published in a more specialized journal

We would like to thank you for the comments. We agree that the theme might be a little specific and could be sent to another journal. However, we decided for plos one since it is one of the best peer review open journals. In addition, many journal would reject this paper since it is not appealing to present negative findings, and we thought than that since plos one does not discriminate papers based on the results it would be the best home for our paper. 

The original trial (Blattman C, et al. Reducing crime and violence: Experimental evidence from cognitive behavioral therapy in Liberia. Am Econ Rev. 2017;107: 1165–1206.) is not available in open acces. The methodology is not described but is directed to the original article by Battman, of which there is no free access, so the reader of plos one cannot access important information for the follow-up of this article.

Thank you for the comment. This is important, and we agree with the reviewer. we would like to say that there is an open access repository were you can find full details of the study. We did not refer to it since normally only peer reviewed references are allowed. But given the importance we will include in the manuscript a link to the source of the full information. 

The only figure that shows the manuscript is a copy of the original article with the 4 arms of the essay. it is not clear if there has been a subsequent intervention with only the two arms of the trial used or not...not detailed at all. There is no evidence of losses or dropouts... wasn't there?.

There were not losses to follow-up, originally, they found all participants, but we did have missing data for some variables. We have made this clear in the manuscript. 

A figure should be detailed with the intervention specific to the two arms of the present study.

We agree with the reviewer and have edited figure one to only show what we are going to analyze and show also how many we did include. 

In the discussion, very poor in the depth of the analysis, the results are compared with another study in which a similar intervention was used in patients with chronic low back pain. Despite the fact that the same authors explain it, CBT is not really comparable in both groups.

Thank you for the comment. We have removed that statement regarding a study that was not comparable to ours. We have also revised the discussion section.

The bibliography should be reviewed, there are citations that do not meet the standards.

We have revised the bibliography, so they all meet the standards.

---

## [Decision Letter · Decision Letter 1]

3 Mar 2023

Combining cash transfers and cognitive behavioral therapy to reduce antisocial behavior in young men: A mediation analysis of a randomized controlled trial in Liberia

PONE-D-22-23021R1

Dear Dr. Loret de Mola,

We’re pleased to inform you that your manuscript has been judged scientifically suitable for publication and will be formally accepted for publication once it meets all outstanding technical requirements.

Kind regards,

Bárbara Oliván-Blázquez, Ph.D.

Academic Editor

PLOS ONE

Additional Editor Comments (optional):

Dear authors,

I have the pleasure to confirm that both reviewers, after a first revision, consider that the manuscript titled Combining cash transfers and cognitive behavioral therapy to reduce antisocial behavior in young men: A mediation analysis of a randomized controlled trial in Liberia, has the neccessary quality for being published in the journal Plos One.

Congratulations!

Reviewers' comments:

Reviewer's Responses to Questions

**Comments to the Author**

1. If the authors have adequately addressed your comments raised in a previous round of review and you feel that this manuscript is now acceptable for publication, you may indicate that here to bypass the “Comments to the Author” section, enter your conflict of interest statement in the “Confidential to Editor” section, and submit your "Accept" recommendation.

Reviewer #1: All comments have been addressed

Reviewer #2: All comments have been addressed

2. Is the manuscript technically sound, and do the data support the conclusions?

Reviewer #1: Yes

Reviewer #2: Yes

3. Has the statistical analysis been performed appropriately and rigorously? 

Reviewer #1: Yes

Reviewer #2: Yes

4. Have the authors made all data underlying the findings in their manuscript fully available?

Reviewer #1: No

Reviewer #2: Yes

5. Is the manuscript presented in an intelligible fashion and written in standard English?

Reviewer #1: Yes

Reviewer #2: Yes

6. Review Comments to the Author

Reviewer #1: (No Response)

Reviewer #2: The authors have substantially improved the manuscript, facilitating access to the original sources, improving the results and discussion. There is a typographical error that should be checked

7. PLOS authors have the option to publish the peer review history of their article (what does this mean?). If published, this will include your full peer review and any attached files.

Reviewer #1: **Yes: **Alejandra Aguilar-Latorre

Reviewer #2: No

---

## [Editor Report · Acceptance letter]

10 Mar 2023

PONE-D-22-23021R1 

 Combining cash transfers and cognitive behavioral therapy to reduce antisocial behavior in young men: A mediation analysis of a randomized controlled trial in Liberia 

Dear Dr. Loret de Mola:

I'm pleased to inform you that your manuscript has been deemed suitable for publication in PLOS ONE. Congratulations! Your manuscript is now with our production department. 

Kind regards, 

on behalf of

Dr. Bárbara Oliván-Blázquez 

Academic Editor

PLOS ONE